



# How methane emission from rice paddy is affected by management practices and region?

Jinyang Wang[1,2], Hiroko Akiyama[3], Kazuyuki Yagi[3], and Xiaoyuan Yan[1]

[1]State Key Laboratory of Soil and Sustainable Agriculture, Institute of Soil Science, Chinese Academy
of Sciences, Nanjing 210008, People's Republic of China

[2]Environment Centre Wales, School of the Environment, Natural Resources and Geography, Bangor
University, Bangor LL57 2UW, United Kingdom

[3]Institute for Agro-Environmental Sciences, National Agriculture and Food Research Organization, 3-1-
3, Kannondai, Tsukuba, Ibaraki 305-8604, Japan

*Correspondence to*: X.Y. Yan (yanxy@issas.ac.cn)

## Abstract

Rice cultivation has long been known as one of the dominant anthropogenic contributors to methane ($CH_4$) emissions, yet there is still uncertainty when estimating its emissions at the global/regional scale. An increasing number of rice field measurements have been conducted globally, which allow us to assess the major variables controlling $CH_4$ emissions and develop the region- and country-specific emission factors (EFs). Results of our statistical analysis shown that the $CH_4$ flux from rice fields were closely related to organic amendment, water regime during and before the rice-growing season, soil properties and climate. The average $CH_4$ flux from fields with single and multiple drainages were 71% and 55% of that from continuously flooded rice fields. The $CH_4$ flux from fields that were flooded in the previous season were 2.4 and 2.7 times that from fields previously drained for a short and long season. Contrary to the previously reported optimum soil pH of around neutrality, paddy soils with pH of 5.0–5.5 gave the maximum $CH_4$ emission. Rice straw applied at 6 t $ha^{-1}$ shortly before rice transplanting can increase $CH_4$ emission by 3.2 times, while it increases $CH_4$ emission by only 1.6 times when applied in the previous season. The default EF was estimated to 1.19 kg $CH_4$ $ha^{-1}d^{-1}$ with a 95% confidence interval of 0.80 to 1.76 kg $CH_4$ $ha^{-1}d^{-1}$ for continuously flooded rice fields without organic amendment and with a preseason water status of short drainage. The default EFs at sub-regional and country levels were also estimated. We conclude that these default EFs and scaling factors can be used to develop national or regional emission inventories.



## 1 Introduction

Atmospheric methane ($CH_4$) is an important greenhouse gas (GHG), and its global mean concentration has increased by a factor of 2.5 since the pre-industrial era (Dlugokencky et al., 2011). It has long been recognized that rice cultivation is one of the dominant anthropogenic contributors to $CH_4$ emissions (Ciais et al., 2013; Koyama, 1963). Over the last century, the observed expansion of rice fields was the dominant factor for the increase of global $CH_4$ emissions from rice cultivation (Fuller et al., 2011; Zhang et al.,

2016). Owing to the increasing area of rice grown globally, the increase in $CH_4$ emission is expected to continue in the near future (EPA, 2012; FAO, 2016).

While the total global $CH_4$ source is relatively well known, the strength of each source component and their trends remains uncertain. Although substantial progress has been made in estimating $CH_4$ emissions from global rice fields over the last three decades, large discrepancies in magnitude exist among

various studies (range: 20.8 to 170 Tg $CH_4$ $yr^{-1}$; Cicerone and Oremland, 1988; EPA, 2012; Frankenberg, 2005; Neue et al., 1990; Yan et al., 2009). Previous studies have shown that the magnitude of estimated $CH_4$ emissions from rice cultivation turned out a downward trend, suggesting that the estimated accuracy has been improved. In general, the estimations from top-down approaches (31−112 Tg $CH_4$ $yr^{-1}$) (IPCC, 2007) were much higher than those from both inventory (25.6−41.7 Tg $CH_4$ $yr^{-1}$) (EPA, 2012; FAO,

2016; Yan et al., 2009) and bottom-up (18.3−44.9 Tg $CH_4$ $yr^{-1}$) approaches (Ito and Inatomi, 2012; Spahni et al., 2011; Zhang et al., 2016). For example, time series (1990-2012) estimation of $CH_4$ emissions from global rice fields by Emission Database for Global Atmospheric Research (EDGAR) (http://edgar.jrc.ec.europa.eu/part_CH4.php) was higher than those reported by FAO (http://faostat3.fao.org/home/E) and Environmental Protection Agency

(http://epa.gov/climatechange/ghgemissions/gases/ch4.html). Such discrepancies may be the result of the higher estimation of prior information on either rice field distribution or the estimated $CH_4$ emissions being used in the top-down studies. Furthermore, anthropogenic sources are dominant over natural sources to global $CH_4$ emissions in the top-down studies, while they are of the same magnitude in the bottom-up models and inventories (Ciais et al., 2013).

For national-level reporting of GHG emissions to the United Nations Framework Convention on Climate Change (UNFCCC), a range of methodological approaches is endorsed in IPCC guidelines (i.e., 1996, 2000, 2003, 2006), which are specified under inventory- (i.e., Tier 1 and Tier 2) or model-based approaches (Tier 3). Accordingly, a range of approaches at various Tiers is applied in the UNFCCC GHG



dataset, which provides emissions data communicated by member countries (UNFCCC, 2017). At the
country level, the inventory-based approach is often used for estimating $CH_4$ emissions from rice fields.
For most countries (i.e., South and Southeast Asian countries), either the Tier 1 or Tier 2 method has been
used to compute $CH_4$ emissions from rice fields in their national communications. Although the Tier 2
method requires more specific national values, country-specific emission factors (EFs) and/or scaling
factors (SFs) obtained therein are simply adjusted based on those default values used in the Tier 1 method.
By contrast, the Tier 3 method to date has been used by a few countries to estimate $CH_4$ emissions from
rice cultivation in their national GHG inventory reports, including China, United States, Japan and India
(UNFCCC, 2017). Moreover, to estimate the $CH_4$ emission from rice fields on a global scale, studies
using the IPCC 2006 guidelines showed comparable results (EPA, 2017; FAO, 2016; Tubiello et al., 2013;
Yan et al., 2009). Thus, these findings indicate that the inventory-based methods are useful to provide a
reliable estimation of $CH_4$ emission from rice fields.

The net $CH_4$ flux is determined by both the production from methanogens and the consumption from
methanotrophs (Conrad, 2007). Previous studies have shown that $CH_4$ emissions from rice fields were
influenced by water management (Wang et al., 2012; Zou et al., 2005), nitrogen (N) fertilizer use (Banger
et al., 2012), organic input (Feng et al., 2013; Wang et al., 2013) and rice varieties (Jiang et al., 2017;
Watanabe et al., 1995). By a statistical analysis of a large data set of field measurements, Yan et al. (2005)
revealed that the primary factors that control $CH_4$ emission were organic amendment, agroecological zone,
water regimes during and before the rice-growing season and soil properties. These factors have been
accounted for in the current IPCC guidelines, where EFs and SFs for $CH_4$ emission from rice cultivation
were revised accordingly (Lasco et al., 2006).

Since Yan et al. (2005) was published, there are numerous field measurements in Asian countries.
For the rest of the world, many studies to date have investigated the impact of various factors on $CH_4$
emission from rice fields, while they were not included in the previous analysis (Yan et al., 2005).
Through an updated analysis, therefore, the objectives of this study were (1) to reassess the impacts of
major variables controlling $CH_4$ emission from rice fields, and (2) to develop the region- and country-
specific EFs for which sufficient number of measurements are available.





## 2 Materials and Methods

2.1 Data compilation

Since 2004, there are a large body of field measurements of $CH_4$ emission from rice fields across the world. With a cut-off date on June 31, 2017, the data set of (Yan et al., 2005) was updated and expanded

to include all available observations of $CH_4$ emission from rice fields in the world. We conducted a comprehensive search of the literature reporting the field measurements of $CH_4$ as described previously (Yan et al., 2005). This included a keyword search using the ISI Web of Science (Thomson Reuters, New York, NY, USA) and Google Scholar (Google, Mountain View, CA, USA). For individual studies, the following documented information were compiled: the average $CH_4$ flux in the rice-growing season,

integrated seasonal emission, water regime during and before the rice-growing season, the timing, type and amount of organic amendment, N fertilization, soil properties (i.e., SOC and soil pH), location, agroecological zone, year, duration and season of measurement. As suggested previously (Yan et al., 2005), hourly or daily flux can be a better index of emission strength than seasonal integrated emission. When the average seasonal $CH_4$ flux was not directly reported, it was thus estimated from integrated

seasonal emissions and measurement period, and *vice versa*. The raw data were either obtained directly from    tables    and    texts    or    extracted    by    digitizing    graphs    using    G3DATA    software (http://www.frantz.fi/software/g3data.php).

As shown in Table 1, the water regime in the rice-growing season was determined as continuous flooding, single drainage, multiple drainage, wet season rainfed, dry season rainfed, or deep water. The

preseason water status was classified as flooded, long drainage, short drainage, two drainage. Note that although we tried our best to judge the water status of rice fields from the papers, the water regimes in both the rice-growing season and preseason still could not be determined for some studies, so a level 'unknown' was assigned. For organic amendments, the materials used in the original papers were classified as compost, farmyard manure, green manure or straw. The timing of rice straw application was

distinguished as on-season or off-season. The amount of organic amendment was recorded directly from the original papers with dry weight for straw and fresh weight for other materials. To account for the spatial variability of $CH_4$ emissions on the global scale, experimental sites were classified into different zones based on their climatic conditions. On the basis of temperature and rainfall differences, rice fields in Asia are placed into seven agroecological zones (AEZs 1-3, 5-8) in the FAO zoning system (IRRI,

2002). Rice fields from regions of Latin America, Europe and United States are grouped into three zones.





Because of the limited availability of information on other properties, only SOC and soil pH as continuous variables are included in our data set. If soil organic matter content rather than SOC was reported, it was converted to SOC using a Bemmelen index value of 0.58. In order to meet the requirement of the statistical model, we excluded these measurements with the absence of available information for

these three continuous variables (SOC, soil pH and the amount of organic amendment). Thus, the final data set consisted of 1089 measurements from 122 rice fields across the world, of which 388 field measurements taken from Asian rice fields since 2004 and 147 field measurements from the rest regions of the world (Data Set S1, Figure 1).

## 2.2 The statistical model for controlling factors

Consistent with our previous study (Yan et al., 2005), a linear mixed model is used to explore the effect of controlling variables on $CH_4$ flux from rice fields. It has been suggested that such a model is suitable for analyzing unbalanced data, that is, data having unequal numbers of observations in the subclasses (Speed et al., 2013). The data set of this study is of this nature, as they were collected simply from non-systematically designed experimental results. Fluxes of $CH_4$ do not fit a normal distribution, they fit a

log-normal distribution. The linear model was used to analyze the log-transformed data of $CH_4$ flux as follows:

$$\ln(flux) = constant + a \times \ln(SOC) + pH_h + PW_i + WR_j + CL_k + OM_l \times \ln(1 + AOM_l),$$

(1)

where flux is the average $CH_4$ flux during the rice-growing season; SOC and $a$ represent SOC content

and its effect, respectively; $pH_h$, $PW_i$, $WR_j$, $CL_k$, and $OM_l$ represent the effects of soil pH, preseason water status, water regime in the rice-growing season, climate and organic amendment, respectively; $AOM_l$ is the amount of organic amendment in t ha$^{-1}$. In this model soil pH was treated as a categorical variable and grouped into the following classes ($h$): <4.5, 4.5-5.0, 5.0-5.5, 5.5-6.0, 6.0-6.5, 6.5-7.0, 7.0-7.5, 7.5-8.0 and ≥8.0. For other categorical variables, their corresponding sublevels ($i, j, k, l$) are shown in Table

140    1.

The last part of Eqn (1) reflects the effect of organic amendment on $CH_4$ flux from rice fields, which is an interaction of the type and amount of organic materials used. To ameliorate the problem induced by the assumption that the treatment without organic amendment is the result of each type of organic material at zero application rate, we weighted the residual of observations with organic amendment as 1 and those



without as 0.2. The effects of the controlling variables on CH₄ flux were computed by fitting Eqn (1) to

field observations using the SPSS Mixed Model procedure (V24.0, SPSS Inc., Chicago, IL, USA).

2.3 Developing global and region-/country-specific emission factors

The estimated effects of various variables were used to derive a default EF. In the model, the CH₄

emission from rice fields is a combination of the effects of SOC and pH values, preseason water status,

water regime in the rice-growing season, organic amendment and climate. An assumption was made to

provide a default EF, that is, all observations in the data set to have a water regime of continuous flooding,

a preseason water status of short drainage and no organic amendments, while keeping other conditions as

stated in the original papers. Then, we derived a default EF for continuously flooded rice fields with a

preseason water status of short drainage and without organic amendments using Eqn (2):

$$EF = e^{constant} \times \left(\frac{1}{n}\sum_{i=1}^{n} SOC_i^a \times e^{pH_i} \times e^{CL_i}\right) \times e^{PW_{short\,draiange}} \times e^{WR_{continuous\,flooding}} \quad ,$$

(2)

where '*constant*' and '*a*' are the values estimated in Eqn (1), *n* is the total number of observations in the

data set, $pH_i$ and $CL_i$ are the effects of pH and climate of the *i*th observation, respectively; $PW_{short\,drainage}$

and $WR_{continuous\,flooding}$ are the effects of preseason short drainage and continuous flooding in the rice season,

respectively.

In the 2006 IPCC guidelines, the Tier 1 method is meant to be applied to countries in which CH₄

emissions from rice cultivation are not a key category or for which country-specific EFs do not exist

(Lasco et al., 2006). Thus, in the Tier 2 method the use of country-specific EFs is encouraged. To take

advantage of the estimated effects of various variables at the global level, region- or country-specific EFs

can be developed for some regions where sufficient number of CH₄ emission measurements from rice

fields to date are available.

**3 Results and Discussion**

3.1 The advantages of the statistical model

An advantage of this linear mixed model is that it can handle many variables together, and makes use of

the large number of unsystematic field measurements (Jørgensen and Fath, 2001; Yan et al., 2005).

Results of our previous modeling analysis (Yan et al., 2005) have been adopted by the 2006 IPCC





guidelines as the inventory-based (i.e., Tier 1 and 2 methods) approaches in which a baseline default EF and various SFs are estimated (Lasco et al., 2006). Moreover, the estimation of global $CH_4$ inventory from rice cultivation using the 2006 IPCC guidelines (Yan et al., 2009) is comparable to other estimations

(EDGAR, 2017). Although empirical or mechanistic models are also encouraged to be used for estimating $CH_4$ emissions during rice cultivation, we are aware of that only a few countries such as China (CH4MOD) (Huang et al., 2006), United States (DAYCENT) (Cheng et al., 2014) and Japan (DNDC-Rice) (Katayanagi et al., 2016), used this approach in their submitted national communications to the Conference of the Parties (UNFCCC, 2017). For most countries, either the default or country-specific

EFs (if available) are used to develop their national inventories of $CH_4$ emission from rice fields. Thus, it is still necessary to develop the global default or region-/country-specific EFs with the statistical modeling.

Variables considered in the present model are SOC, soil pH, preseason water status, water regime in the rice-growing season, organic amendment and climate (Table 2). Although the $CH_4$ emission from rice fields can also be influenced by many other factors such as other soil properties, N fertilization, rice

cultivar (Aulakh et al., 2001; Banger et al., 2012; Conrad, 2007), those factors are not considered here because there are contradictory reports on their effects, or very limited information on the variables *per se* is available. For instance, to date there is no single consensus on N fertilization impacts on $CH_4$ emissions from rice fields. It is likely attributed to the highly complex nature of the effect of N fertilizer on $CH_4$ emission which can strongly interact with other factors such as the amount and type of N fertilizer

and water regime (Banger et al., 2012). Further, very few countries (i.e., Indonesia) considered the effects of soil type and rice cultivar on $CH_4$ emission from rice fields in their national communications. There is also large inter-annual variability in $CH_4$ flux (Shang et al., 2011; Wang et al., 2012), which cannot be reflected in the current model. Nevertheless, the selected variables in the current model can account for 50% of the variability in $CH_4$ emissions on the global scale.

3.2 Effects of controlling variables

At the global scale, SOC and soil pH are the soil properties in controlling $CH_4$ emission from rice fields, while the contribution of SOC to the variance is the smallest among all variables considered here ($F_{(1, 3391)}$ = 39.8, $P < 0.0001$; Table 2). This finding may indicate that the controlling effect of SOC on $CH_4$ emission from rice fields on a global scale may be overweighed by other variables (i.e., organic amendments). For

example, although a recent synthesis by Banger et al. (2012) showed a positive but weaker ($R^2 = 0.21$) relationship between SOC content and the $CH_4$ flux, they did not consider $CH_4$ emissions from rice fields



with organic amendments. Further, in a Chinese double rice-cropping system, the long-term (*c.* 11 yr) organic amendment-induced increase in SOC may be responsible for the observed significant correlation between SOC and $CH_4$ emission (Shang et al., 2011). Previous studies also suggested that the content of readily mineralizable carbon rather than SOC was significantly correlated with $CH_4$ emission from rice fields (Yagi and Minami, 1990). Thus, we believe that a weak relation between SOC and $CH_4$ emission at the global scale can be largely attributed to the dominant factors controlling $CH_4$ emissions are labile C substrates derived from inherent and exogenous sources (Wang et al., 2013; Yagi and Minami, 1990).

The effect of soil pH on controlling $CH_4$ emission from rice fields is not monotonic ($F_{(8, 3391)} = 75.3$, $P < 0.0001$; Table 2), which is consistent with the previous results (Yan et al., 2005). It is often accepted that $CH_4$ production under anoxic conditions is very sensitive to variations in soil pH, as the activity of methanogens is usually optimum around neutrality or under slightly alkaline conditions (Aulakh et al., 2001; Garcia et al., 2000). However, soils with a pH of 5.0-5.5 showed a much higher emission than other soils, which corroborates the observed relationship between soil pH and $CH_4$ emission in Indonesian rice fields (Yan et al., 2003). The largest effects of soil pH below 4.5 may not be reliable because of limited observations from only two studies with distinct water regimes, soil properties and organic amendments. Given that methanogens and methanotrophs are tolerant to pH variations in soil (Dunfield et al., 1993), and $CH_4$ emission is the result of its production, consumption and transfer in soil to the atmosphere (Conrad, 2007), we suppose that it is not soil pH itself, but some other soil properties or microbial activities correlating with soil pH that control these processes. Thus, we conclude that such correlation between soil pH and $CH_4$ emission at the global scale may be reasonable.

As expected, water regime in the rice-growing season is a main factor controlling $CH_4$ emission from rice fields ($F_{(6, 3391)} = 80.5$, $P < 0.0001$; Table 2). Relative to continuous flooding, the average seasonal $CH_4$ flux in the rice-growing season can be reduced by 29% and 45% by single and multiple drainage, respectively (Table 3). In the updated data set, the magnitude of reducing $CH_4$ emission following single drainage is smaller than previous results (Yan et al., 2005). This may be due not only to *c.* 3 times increment of available observations (Data Set S1) but also to inevitably confusion for identifying water regime from different studies. The average $CH_4$ fluxes from wet-season and dry-season rainfed rice fields are 54% and 16%, respectively, of that from continuously flooded fields, lower than the IPCC values of 80% and 40% for flood-prone rainfed and drought-prone rainfed rice fields, respectively (IPCC, 1997). Compared with the previous results (Yan et al., 2005), the greater average $CH_4$ flux from wet-season rice



fields is mainly attributed to the observed high fluxes from rainfed rice fields in Thailand and India (Kaewpradit et al., 2008; Kantachote et al., 2016; Rath et al., 1999). Yet, the $CH_4$ flux from deep water rice, only 6% of that from continuously flooded rice fields, remains less reliable due to the lack of 235 sufficiently observational data in the current analysis.

This statistical model clearly identifies the effects of preseason water status on $CH_4$ emission in the rice-growing season ($F_{(4, 3391)}$ = 94.9, $P$ < 0.0001; Table 2). A negative correlation was found between $CH_4$ emission and the drainage period before the rice season, such that the average $CH_4$ flux from a rice field that was flooded in the previous season is 2.4−4.1 times as high as that from fields that experienced 240 different durations of drained season (Table 3). As shown in Table 1, preseason water status is determined mainly by the crop rotation system, except in rice fields that are flooded during the fallow season. This effect of preseason water condition can explain some of the regional and seasonal differences of $CH_4$ emissions from rice fields, and suggests that cultivating rice alternately with upland crops considerably may contribute to mitigating $CH_4$ emissions from rice fields.

Among all the selected variables, the effect of organic amendment is the biggest ($F_{(5, 3391)}$ = 181.5, $P$ < 0.0001), suggesting that the use of organic materials is the main variable controlling $CH_4$ emissions from rice fields. Among all the organic materials, straw used on-season shows the strongest stimulating effect on $CH_4$ emission, followed by green manure. Such a difference may be attributed not only to the decomposition but also to the different moisture contents of organic materials recorded in the literature 250 (Table 1). If rice straw applied at a rate of 6 t ha$^{-1}$ (dry weight) before rice transplanting, the $CH_4$ emission was 3.2 times that from fields without any organic amendment (Figure 2). However, when this amount of rice straw is incorporated to soil immediately after harvest in the previous year and left unflooded, the stimulating effect on $CH_4$ emission is only 1.6 times. This indicates that straw applied off-season is an effective way to reduce $CH_4$ emission from rice fields. The stimulating effects of compost and farmyard 255 manure are comparable to that of rice straw applied off-season.

Although the climate affect $CH_4$ emission significantly ($F_{(9, 3391)}$ = 52.4, $P$ < 0.0001), its contribution to the variance is smaller than other factors considered in the model. It is likely because the model considers soil properties, water regime during and before the rice-growing season, which partially reflect the effect of climate. As shown in Table 2, the highest effect of AEZ 1 with an extremely large variability 260 is still unreliable, because there is no new data added in our data set. The higher $CH_4$ emissions can be



identified clearly for AEZ 2 and 6 and Europe as the 95% confidence intervals of their effects are not overlapped with others.

3.3 Development of region- or country-specific emission factors

Globally, for continuously flooded rice fields with preseason water status of short drainage without
organic amendment, the EF is estimated to be 1.19 kg $CH_4$ $ha^{-1}d^{-1}$ with an error range of 0.80-1.76 kg $CH_4$ $ha^{-1}d^{-1}$ (Table 4). This estimate is lower than 1.30 kg $CH_4$ $ha^{-1}d^{-1}$ of Yan et al. (2005) which had relatively large variation (0.80-2.20 kg $CH_4$ $ha^{-1}d^{-1}$). Such a difference could be mainly attributed to the number of field measurements in the present data set about two times greater than in their study. As shown in Table 4, we estimated the region- or country-specific EFs for which sufficient number of $CH_4$ emission
measurements from rice fields are available.

*East Asia* Approximately 90% of the world's rice fields are located in Asia, of which 23% are occurred in East Asia (FAO, 2016). In this data set, about half of $CH_4$ emission measurements are compiled from this region (Figure 1; Data Set S1). The region-specific EF for East Asia is estimated to 1.32 kg $CH_4$ $ha^{-1}d^{-1}$, and there are differences in the country-specific EF in the order South Korea > China > Japan
(Table 4). For China, as the biggest rice producer in the world, there is a growing body of $CH_4$ emission measurements from rice fields since the late 1980s (Figure 1). We collated 388 field observations conducted on more than 40 sites in China, which allowed us to make a relatively reliable estimate for the country-specific EF. Although the EF of 1.30 kg $CH_4$ $ha^{-1}d^{-1}$ (error range: 0.84-1.87 kg $CH_4$ $ha^{-1}d^{-1}$) is the same to the IPCC latest default EF, its variability is smaller than the latter one with an error range of
0.80-2.20 kg $CH_4$ $ha^{-1}d^{-1}$ (Lasco et al., 2006). This is supported by the evidence that the $CH_4$ emission from Chinese rice fields estimated using the Tier 1 method in the 2006 IPCC guidelines or country-specific EF are almost identical (7.22-8.64 Tg $yr^{-1}$) (Yan et al., 2003, 2009). Even though the estimate of $CH_4$ emission beyond the scope of this study, we believe, to some extent, that developing and using country-specific EF should be a promising approach for national $CH_4$ inventory. For example, using the
process-based model (CH4MOD) and empirical method to account for different EFs in various rice ecosystems, $CH_4$ emission from rice cultivation in year 2012 is estimated to be 8.46 Tg $yr^{-1}$ in China's First Biennial Update Report (BUR) to its National Communications (NDRC of China, 2016). These estimates accounting for various EFs under different conditions falls into the range of 4.98-14.19 Tg $yr^{-1}$ from other reports (EDGAR, 2017; EPA, 2017; FAO, 2016).



In the latest National Communication under the Convention of Japan, country-specific EFs for rice

fields under different water regimes during the rice-growing season were estimated using the DNDC-

Rice model (Katayanagi et al., 2016; MoE of Japan, 2017). For comparison, the length of the single rice

season in East Asia is assumed to be 130 days (Yan et al., 2005), we found that our estimate (1.06 kg $CH_4$

$ha^{-1}d^{-1}$) falls into the range of the model-derived EF ranging from 0.06 to 1.79 kg $CH_4$ $ha^{-1}d^{-1}$ for

continuously flooded rice fields without organic amendment across Japan (Katayanagi et al., 2016).

Likewise, Yan et al. (2009) using the Tier 1 method estimated the $CH_4$ emission in year 2000 from

Japanese rice fields to be 407 Tg $yr^{-1}$, which is lower than the 510 Tg $yr^{-1}$ in their latest report (MoE of

Japan, 2017). We argued that such a discrepancy may be primarily related to different classifications for

intermittently flooded (i.e., single drainage vs. multiple drainage) and type and amount of organic

amendment used in their estimations. As such, we believe that when reliable information regarding water

management and organic amendment becomes available, there is still merit in using the current country-

specific EF for national $CH_4$ emission from rice cultivation. Also, it could be the case for South Korea,

because $CH_4$ emission estimate using the Tier 1 method appears comparable to that of their National

Communications (Yan et al., 2009).

*South Asia* The rice harvest area of countries in South Asia accounts for 42% of the Asian total rice

harvest for the year 2010 (FAO, 2016). India is currently thought to have the second largest $CH_4$ emissions

from rice cultivation in the world. In the present study, the estimated EF of $CH_4$ from Indian rice fields

was 0.85 kg $CH_4$ $ha^{-1}d^{-1}$ (error range: 0.57-1.25 kg $CH_4$ $ha^{-1}d^{-1}$). We find that our estimate agrees with

the overall average of $0.59 \pm 0.35$ kg $CH_4$ $ha^{-1}d^{-1}$ ($\pm$ SD, the length of rice season is assumed to 125 days),

which was used for $CH_4$ emission inventory from Indian rice cultivation (MoEFCC of India, 2015).

Interestingly, if the SFs (Table 3) are applied for subcategories of water regime during the rice-growing

season as in the Tier 1 method (Lasco et al., 2006), our estimates for irrigated rice fields are almost

identical to those of Manjunath et al. (2009), which has been consistently used in their national $CH_4$

inventory. By contrast, the values for rainfed and deep water fields are greatly underestimated. This

discrepency is primarily due to the fact that peer-reviewed studies from India are only considered in our

current data set, while 471 observations collected from farmer's fields over India are used by Manjunath

et al. (2009). The aforementioned limited data points from wet and dry-season rainfed rice fields may also

lead to biased estimates, despite about half of rice cultivation under rainfed conditions in India's first

BUR. This indcates that further available observations of $CH_4$ emissions from rainfed and deep water rice

fields are required to improve our statistical estimates.



For Bangladesh, albeit based on one study, the estimated EF (0.97 kg $CH_4$ $ha^{-1}d^{-1}$) of $CH_4$ emission from rice fields is for the first time available. Previous studies often used an EF vaule from neighboring countrires for $CH_4$ emission estimates from rice cultivation (FAO, 2016; Manjunath et al., 2014; Yan et al., 2003, 2009). Interestingly, our estimate is similar to the seasonally integrated EF value of 10 g $CH_4$

$m^{-2}$ used in their national communications (MoEF of Bangladesh, 2012) or other reports (FAO, 2016). Furthermore, previous studies have shown that the national $CH_4$ estimates were comparable when using the EF from their neighboring countries (Manjunath et al., 2014; Yan et al., 2009). Thus, either the region (0.85 kg $CH_4$ $ha^{-1}d^{-1}$) or these country-specific EFs could be used for $CH_4$ emission estimates from the rest countries of South Asia, *viz*., Pakistan, Sir Lanka and Nepal where direct measurements to date are

not available or insufficient (Table 4).

*Southeast Asia* In Southeast Asia, the total $CH_4$ emission from rice cultivation accounted for 21.5% of the world total (Yan et al., 2009). The EF of 1.22 kg $CH_4$ $ha^{-1}d^{-1}$ for this region is close to the global default value but differs among countries (Table 4). Country-specific EFs (kg $CH_4$ $ha^{-1}d^{-1}$) for each country are estimated to be *viz*. Indonesia (1.18), Philippines (0.60) and Viet Nam (1.13). For Indonesia,

the EF with an average of 160.9 kg $CH_4$ $ha^{-1}season^{-1}$ is used for $CH_4$ inventory from rice cultivation, despite the existence of large variation in field measurements (6.7-798.6 kg $CH_4$ $ha^{-1}season^{-1}$) (MoEF of Indonesia, 2015). Because the length of the rice season in Southeast Asia countries varies from 99 to 115 days, our estimate is close to the default EF used in their first BUR. For Philippines, our estimate is much lower than 3.46 kg $CH_4$ $ha^{-1}d^{-1}$ estimated by Yan *et al.* (2003) based on observations from only two sites.

Using the Tier 1 method in the 2006 IPCC guidelines, Yan et al. (2009) found the estimates of $CH_4$ emission from Philippines and Viet Nam rice fields agreed reasonably well with the values reported in their National Communications. The larger EFs estimated for Thailand and Cambodia (data not shown) have big uncertainties because they are essentially developed from very limited observations.

*Americas* Rice cultivation in Brazil and United States accounts for about 60% of the total in Americas

(FAO, 2016). In our data set, there are only three countries from this region having available measurements which allowed us to make country-specific EF estimates (Table 4). The country-specific EFs are estimated to be 0.65, 1.62 and 0.80 kg $CH_4$ $ha^{-1}d^{-1}$ for United States, Brazil and Uruguay, respectively. By contrast, the assigned values of seasonally integrated EF for the corresponding countries are 35, 6.5 and 28 g $CH_4$ $m^{-2}$ in FAOSTAT emission database (FAO, 2016). Using the IPCC Tier 1

method, the $CH_4$ emission estimate for these countries tends to be lower than that of their national



inventory reports (NIRs), suggesting the importance of country-specific EF as differential conditions for rice cultivation being considered. For example, in the United States's latest NIR, there is an approximately 25% increase in $CH_4$ emission from rice cultivation relative to the previous estimates (EPA, 2017). This change could be resulted from unified continuous flooding in the rice season and the impact of winter

flooding considered in the IPCC Tier 3 method (DAYCENT model). Thus, in a previous study (Yan et al., 2009), the underestimated $CH_4$ emission using the IPCC Tier 1 method for United States can be explained by different assumptions made for water regimes of rice cultivation. Nevertheless, our results should be treated with caution as very limited observations available for these countries.

*Europe* As the major rice cultivated countries in Europe, the country-specific EFs for Italy and Spain are

estimated to be 1.66 and 1.13 kg $CH_4$ $ha^{-1}d^{-1}$, respectively (Table 4). However, a seasonally integrated EF of 50.4 g $CH_4$ $m^{-2}$ is assigned for these two countries in FAOSTAT emission database (FAO, 2016), which is far higher than our estimates as well the values used in their NIRs. In the Italy's NIR, the EFs for continuously flooded fields without organic amendments for single and multiple drainage are 2.0 and 2.7 kg $CH_4$ $ha^{-1}d^{-1}$, respectively. It is interesting to note that these values contradict our expectation that

the $CH_4$ emission should be lower from rice fields with multiple than single drainage (Table 3). A possible reason for this is that they are based on experimental measurements from different rice field studies in Italy (Leip et al., 2002; Meijide et al., 2011). In the latest NIR of Spain, the global default EF (1.30 kg $CH_4$ $ha^{-1}d^{-1}$) is used for $CH_4$ emission estimate from rice cultivation, which is close to our estimate.

## 4 Conclusions

In this study, we present the update of the findings of Yan et al. (2005) through extending the database of $CH_4$ emission from global rice fields. In the statistical model, those selected variables having significant effects on $CH_4$ emission from global rice fields agree well with results of the previous analysis. In the updated data set, the estimated values of default EF and SFs have changed in some cases; for instance, the average $CH_4$ fluxes from rice fields with single drainage was 71% rather than 58% of that

from continuously flooded rice fields. More importantly, not only the global default EF is updated, but also the region- or country-specific EFs are for the first time developed for countries where sufficient number of $CH_4$ emission measurements from rice fields are available. Thus, these default EFs and SFs for different water regimes and organic amendments can be used to develop national or regional emission inventories.



**Acknowledgments**

Jinyang Wang acknowledges the European Commission under Horizon 2020 for support by a Marie
Skłodowska-Curie Actions COFUND Fellowship (663830-BU-048), and the financial support provided
by the Welsh Government and Higher Education Funding Council for Wales through the Sêr Cymru
National Research Network for Low Carbon, Energy and Environment.





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



**Figure 1.** Global distribution of field experiments measuring CH$_4$ flux from rice fields. The circle and

triangle indicate experimental sites newly added in this study and included in Yan et al. (2005),

respectively.

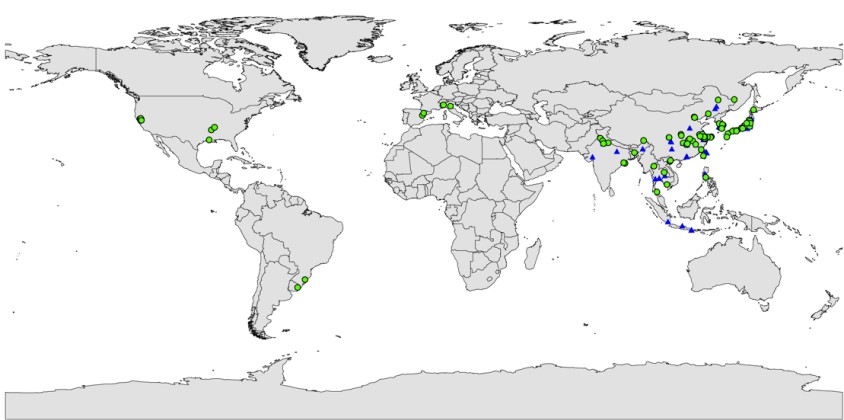




**Figure 2.** Simulated effect of different organic amendments on $CH_4$ emission from rice fields. The $CH_4$

flux for the field without any organic amendment is assumed to be 1.

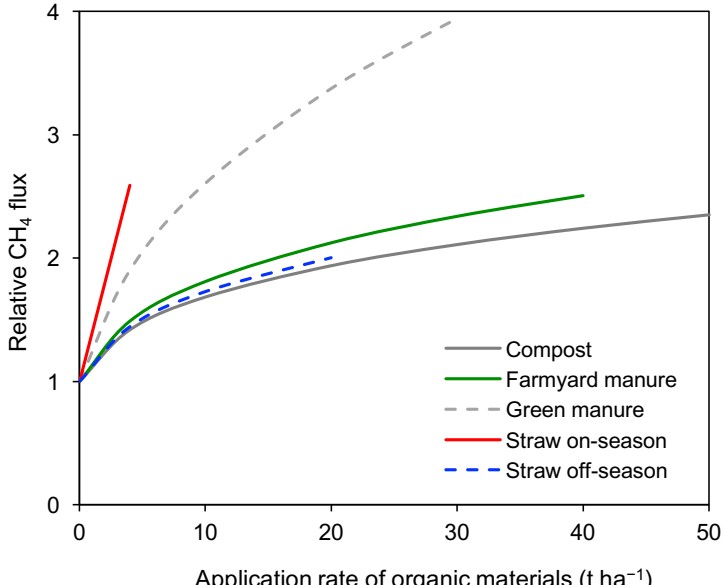




**Table 1.** Description of the selected variables controlling on the CH$_4$ emission from rice fields

| Variables | Description |
| --- | --- |
| Preseason water status | |
| Flooded | Permanently flooded rice fields are assumed to have a preseason water regime of 'flooded'. Late rice in China is usually planted immediately after early rice on the same field and is therefore regarded as having a preseason water regime of 'flooded'. |
| Long drainage | If rice is planted once a year and the field is not flooded in the non-rice growing season, the preseason water regime is classified as long drainage. |
| Short drainage | Rice is planted more than once a year, but there is more than one month fallow time between the two seasons, short drainage is usually taken as preseason drainage. |
| Two drainage | For measurements conducted on rice fields that are preceded by two upland crops or an upland crop and a drained fallow season, the preseason water of such experiments is classified as two drainage. |
| Water regime in the rice-growing season | |
| Continuous flooding | Rice is cultivated under continuously flooded condition but sometimes an end-season drainage before rice harvest included. |
| Single drainage | One mid-season drainage and an end-season drainage are adopted over the entire rice-growing season. |
| Multiple drainage | It refers to the water regime is called 'intermittent irrigation' but the number of drainages was not clear. Alternate wetting and drying (AWD) is included in multiple drainage. |
| Rainfed, wet season | Rice cultivation rely on rainfall for water, in this case the field is flood prone during the rice-growing season. |
| Rainfed, dry season | Rice cultivation rely on rainfall for water, in this case the field is drought prone during the rice-growing season. |
| Deep water | Rice grown in flooded conditions with water depth more than 50 cm deep. |
| Organic amendment | |



| Straw on-season | Straw applied just before rice transplanting as on-season; straw that is left on the soil surface in the fallow season and incorporated into the soil before the next rice transplanting is also categorized as on-season. The amount of straw return is expressed in dry weight. |
|---|---|
| Straw off-season | Straw incorporated into soils in the previous season (upland crop or fallow) is categorized as off-season. The amount of straw return is expressed in dry weight. |
| Compost, farmyard manure, green manure | The amount of organic materials is expressed in fresh weight. |
| Agroecological zone | |
| AEZ 1 | Warm arid and semiarid tropics |
| AEZ 2 | Warm subhumid tropics |
| AEZ 3 | Warm humid tropics |
| AEZ 5 | Warm arid and semiarid subtropics with summer rainfall |
| AEZ 6 | Warm subhumid subtropics with summer rainfall |
| AEZ 7 | Warm/cool humid subtropics with summer rainfall |
| AEZ 8 | Cool subtropics with summer rainfall |



**Table 2.** Statistical results for fixed effects obtained by fitting the model to the observed log-transformed $CH_4$ fluxes (mg $CH_4$ m$^{-2}$h$^{-1}$)

| Effect | Estimate | Standard error | df | $t$-value | $P$-value | 95% confidence interval | |
| --- | --- | --- | --- | --- | --- | --- | --- |
| | | | | | | Lower | Upper |
| Constant | -0.478 | 0.171 | 3391 | -2.79 | 0.005 | -0.814 | -0.142 |
| SOC | 0.190 | 0.030 | 3391 | 6.31 | 0.000 | 0.131 | 0.249 |
| pH | | | | | | | |
|   < 4.5 | 2.045 | 0.210 | 3391 | 9.75 | 0.000 | 1.634 | 2.456 |
|   4.5−5.0 | 1.124 | 0.106 | 3391 | 10.60 | 0.000 | 0.916 | 1.332 |
|   5.0−5.5 | 1.299 | 0.094 | 3391 | 13.88 | 0.000 | 1.116 | 1.483 |
|   5.5−6.0 | 0.825 | 0.091 | 3391 | 9.09 | 0.000 | 0.647 | 1.004 |
|   6.0−6.5 | 0.312 | 0.084 | 3391 | 3.69 | 0.000 | 0.146 | 0.477 |
|   6.5−7.0 | 0.151 | 0.088 | 3391 | 1.73 | 0.085 | -0.021 | 0.323 |
|   7.0−7.5 | 0.181 | 0.097 | 3391 | 1.86 | 0.063 | -0.010 | 0.372 |
|   7.5−8.0 | 0.099 | 0.093 | 3391 | 1.07 | 0.285 | -0.083 | 0.280 |
|   ≥ 8.0 | $0^c$ | | | | | | |
| Preseason water status | | | | | | | |
|   Flooded | 0.763 | 0.064 | 3391 | 11.94 | 0.000 | 0.638 | 0.888 |
|   Long drainage | -0.228 | 0.054 | 3391 | -4.20 | 0.000 | -0.335 | -0.122 |
|   Short drainage | -0.116 | 0.061 | 3391 | -1.90 | 0.058 | -0.237 | 0.004 |
|   Two drainages | -0.648 | 0.184 | 3391 | -3.52 | 0.000 | -1.008 | -0.287 |
|   Unknown | $0^c$ | | | | | | |
| Water regime | | | | | | | |
|   Continuous flooding | 0.851 | 0.138 | 3391 | 6.16 | 0.000 | 0.580 | 1.122 |
|   Deepwater | -1.897 | 0.309 | 3391 | -6.14 | 0.000 | -2.503 | -1.291 |
|   Multiple drainage | 0.247 | 0.142 | 3391 | 1.74 | 0.082 | -0.032 | 0.525 |
|   Single drainage | 0.505 | 0.147 | 3391 | 3.45 | 0.001 | 0.218 | 0.793 |
|   Rainfed, wet season | 0.236 | 0.161 | 3391 | 1.46 | 0.144 | -0.081 | 0.552 |
|   Rainfed, dry season | -0.972 | 0.199 | 3391 | -4.89 | 0.000 | -1.361 | -0.582 |
|   Unknown | $0^c$ | | | | | | |



| | | | | | | | |
|---|---|---|---|---|---|---|---|
| **Organic amendment** | | | | | | | |
| Compost | 0.218 | 0.047 | 3391 | 4.65 | 0.000 | 0.126 | 0.309 |
| Farmyard manure | 0.247 | 0.028 | 3391 | 8.90 | 0.000 | 0.193 | 0.302 |
| Green manure | 0.400 | 0.026 | 3391 | 15.47 | 0.000 | 0.349 | 0.450 |
| Straw on-season[a] | 0.591 | 0.022 | 3391 | 27.49 | 0.000 | 0.549 | 0.633 |
| Straw off-season[a] | 0.228 | 0.036 | 3391 | 6.39 | 0.000 | 0.158 | 0.299 |
| Unknown | 0[c] | | | | | | |
| **Agroecological zone[b]** | | | | | | | |
| AEZ 1 | 1.523 | 0.508 | 3391 | 3.00 | 0.003 | 0.528 | 2.518 |
| AEZ 2 | 1.005 | 0.089 | 3391 | 11.24 | 0.000 | 0.829 | 1.180 |
| AEZ 3 | 0.307 | 0.074 | 3391 | 4.17 | 0.000 | 0.163 | 0.451 |
| AEZ 5 | 0.525 | 0.098 | 3391 | 5.38 | 0.000 | 0.334 | 0.717 |
| AEZ 6 | 1.127 | 0.070 | 3391 | 16.00 | 0.000 | 0.989 | 1.265 |
| AEZ 7 | 0.605 | 0.076 | 3391 | 7.94 | 0.000 | 0.455 | 0.754 |
| AEZ 8 | 0.526 | 0.078 | 3391 | 6.76 | 0.000 | 0.373 | 0.678 |
| South America | 0.403 | 0.150 | 3391 | 2.68 | 0.007 | 0.108 | 0.697 |
| Europe | 1.321 | 0.101 | 3391 | 13.08 | 0.000 | 1.123 | 1.520 |
| North America | 0[c] | | | | | | |

[a]The effect of organic amendment is the interaction of organic material type and application rate (t ha$^{-1}$). Straw on-season indicates straw

applied just before rice season, and straw off-season indicates straw applied in the previous season. Note that rice straw that was left in situ

and incorporated into soil just before the rice season is classified as straw on-season.

[b]Experimental sites are classified as one of the agroecological zones according to the FAO zoning system.

[c]For each categorical variable, the effect of one subclass is set to zero.





**Table 3.** Relative fluxes for different water regimes in the rice-growing season and for different preseason

water statuses

| Variables | Relative flux | 95% confidence interval Lower | Upper |
|---|---|---|---|
| Water regime in rice season | | | |
| Continuously flooded | 1[a] | | |
| Deepwater | 0.06 | 0.03 | 0.12 |
| Multiple drainage | 0.55 | 0.41 | 0.72 |
| Single drainage | 0.71 | 0.53 | 0.94 |
| Rainfed, wet season | 0.54 | 0.39 | 0.74 |
| Rainfed, dry season | 0.16 | 0.11 | 0.24 |
| Preseason water status | | | |
| Short drainage | 1[a] | | |
| Long drainage | 0.89 | 0.80 | 0.99 |
| Two drainages | 0.59 | 0.41 | 0.84 |
| Flooded | 2.41 | 2.13 | 2.73 |

[a]Supposing the fluxes of 'continuously flooded' and 'short drainage' to be 1.



**Table 4.** The regional- and country-specific emission factors for CH$_4$ emission (kg CH$_4$ ha$^{-1}$d$^{-1}$) from flooded rice

fields with a preseason water status of short drainage and without organic amendment

| Region | | Emission factor | 95% confidence interval[a] | | Country | Emission factor | 95% confidence interval[a] | |
|---|---|---|---|---|---|---|---|---|
| | | | Lower | Upper | | | Lower | Upper |
| World | | 1.19 | 0.80 | 1.76 | | | | |
| Asia | East Asia | 1.32 | 0.89 | 1.96 | China | 1.30 | 0.88 | 1.93 |
| | | | | | Japan | 1.06 | 0.72 | 1.56 |
| | | | | | South Korea | 1.83 | 1.24 | 2.71 |
| | South Asia | 0.85 | 0.58 | 1.26 | India | 0.85 | 0.57 | 1.25 |
| | | | | | Bangladesh | 0.97 | 0.65 | 1.43 |
| | Southeast Asia | 1.22 | 0.83 | 1.81 | Philippines | 0.60 | 0.41 | 0.89 |
| | | | | | Viet Nam | 1.13 | 0.76 | 1.67 |
| | | | | | Indonesia | 1.18 | 0.80 | 1.74 |
| America | North America | 0.65 | 0.44 | 0.96 | USA | | | |
| | South America | 1.27 | 0.86 | 1.88 | Brazil | 1.62 | 1.10 | 2.40 |
| | | | | | Uruguay | 0.80 | 0.54 | 1.18 |
| Europe | | 1.56 | 1.06 | 2.31 | Spain | 1.13 | 0.77 | 1.68 |
| | | | | | Italy | 1.66 | 1.12 | 2.46 |

[a]Including the uncertainties of the effects of continuous flooding and preseason water status