# Peer review of "Controlling variables and emission factors of methane from global rice fields"

_Atmospheric Chemistry and Physics, 2018_

## Referee Comment (RC1) · Anonymous Referee #1 · 19 Apr 2018

General comments Rice agriculture is an important source of atmospheric methane (CH4). The estimations of CH4 emission from rice fields on a national or global scale have been relatively well documented by using the inventory-based methods or model-based approaches. Due to more and more field measurements of CH4 emission were available from the monsoon Asian countries and the rest of the world in last ten years, the effect of various factors (management practices like water management, nitrogen (N) fertilizer use, organic input and rice varieties, etc.) on CH4 emission from rice fields would be different in statistics from previous reports. However, no information is available on this issue in global scale. The authors updated the dataset from monsoon Asian countries as described previously (Yan et al., 2005) to over the world (1089 measurements from 122 rice fields across the world) in this study. They reassessed

the impacts of major variables controlling CH4 emission from rice fields and found that water management and organic fertilizer application were the top two controlling variables. They developed the region- and country-specific emission factors and also estimated the default EFs at regional and country levels. Overall, the topic of this work was very important and timely to gain an insight into CH4 emission inventory, which would help to assess regional and national agricultural CH4 budget with low uncertainties. Good job! The manuscript was well written too. I recommend this work to be acceptable after minor revisions for publication in Atmospheric Chemistry and Physics.

Minor comments 1. Abstract Please give more information (e.g., EFs or SFs) about the CH4 emission as affected by the region. In other words, the authors should pay much more attention to the regional CH4 emission or emission factors (EFs) besides the management practices. 2. Materials and Methods - Please show the units for all dependent and independent variables in Eqns (1) and (2). - How to quantify the preseason water status (PW) and water regime (WR) in Eqns (1) and (2)? - What's the difference between OM and AOM in Eqn (1)? - It's hard to figure out what the climate variables are. Do the agroecological zones (AEZ) represent climates? If no climate variables were involved in these two equations, I would suggest deleting the CL but showing AEZ. 3. Results and Discussion - Suggest changing '3.3 Development of region- or country-specific emission factors ' to '3.3 Region- and country-specific emission factors' - Please make further discussion to compare the emission factors in this study with IPCC default emission factors.

---

## Referee Comment (RC2) · Anonymous Referee #2 · 4 Jun 2018

Still some language issues, e.g. title is awkward and could deter readers/interest in the paper, many other sentences have unclear meaning and/or awkward language. Paper would definitely benefit from a thorough editing for clarity and language in general.

Specific issues:

1. the authors already know that ln[SOC] and OMx ln[1 + AOM] will be modeled, but we don't know where that information is from.

2. Not sure that treating pH as categorical variable is at all justified or appropriate. Why was this done? Was pH reported from the different field sites in broad categories, or measured with crude litmus paper or similar? That might be a reason, but still... Authors state that the relationship of pH to emissions is 'not monotonic' but from Ta-

ble 2, I don't see strong enough evidence of that, especially given the questionable shoe-horning into many categories—couldn't the slight deviations from a ranked relationship of pH with emissions simply be error? Did the authors try converting pH to concentrations of H+ ions or otherwise back-log-transforming pH values, or other logical numerical ways to treat this definitely-not-categorical variable? I don't think this statement in lines 213-215, "However, soils with a pH of 5.0-5.5 showed a much higher emission than other soils", is really true. It looks to me like soils with the lowest pH values (below 4.5) had the largest effect on CH4 emissions, and the small blips at 5-5.5 and 7 – 7.5 are not necessarily a big deal. No other literature besides the authors' 2005 paper is cited regarding a more complicated relationship between pH and CH4 emissions to support this idea.

3. How did the authors arrive at the weights for the organic matter additions (.2 and 1)? Not clear why this is needed or justified.

4. The authors state several times that because emissions estimates from different authors' inventory assessments, that this means the results are correct/reliable, e.g. line 70, and lines 173-175 where EDGAR estimates are similar to IPCC 2006. This is a truism, though, because doesn't EDGAR use IPCC 2006 defaults to calculate their emissions estimates?

---

## Author Comment (AC1) · 20 Jun 2018

Answers to Referee 1:

General comments Rice agriculture is an important source of atmospheric methane (CH4). The estimations of CH4 emission from rice fields on a national or global scale have been relatively well documented by using the inventory-based methods or model-based approaches. Due to more and more field measurements of CH4 emission were available from the monsoon Asian countries and the rest of the world in last ten years, the effect of various factors (management practices like water management, nitrogen (N) fertilizer use, organic input and rice varieties, etc.) on CH4 emission from rice fields would be different in statistics from previous reports. However, no information is

available on this issue in global scale. The authors updated the dataset from monsoon Asian countries as described previously (Yan et al., 2005) to over the world (1089 measurements from 122 rice fields across the world) in this study. They reassessed the impacts of major variables controlling CH4 emission from rice fields and found that water management and organic fertilizer application were the top two controlling variables. They developed the region- and country-specific emission factors and also estimated the default EFs at regional and country levels. Overall, the topic of this work was very important and timely to gain an insight into CH4 emission inventory, which would help to assess regional and national agricultural CH4 budget with low uncertainties. Good job! The manuscript was well written too. I recommend this work to be acceptable after minor revisions for publication in Atmospheric Chemistry and Physics.

Answer: We would like to thank referee 1 for his/her positive and critical comments on our work. We are glad that referee 1 recognized the importance of our work and we would like to take the opportunity to address concerns of referee 1.

Minor comments 1. Abstract Please give more information (e.g., EFs or SFs) about the CH4 emission as affected by the region. In other words, the authors should pay much more attention to the regional CH4 emission or emission factors (EFs) besides the management practices.

Answer: We followed this suggestion. Results of organic amendment and global or regional emission factors of CH4 were added in the abstract.

2. Materials and Methods - Please show the units for all dependent and independent variables in Eqns (1) and (2). - How to quantify the preseason water status (PW) and water regime (WR) in Eqns (1) and (2)? - What's the difference between OM and AOM in Eqn (1)? - It's hard to figure out what the climate variables are. Do the agroecological zones (AEZ) represent climates? If no climate variables were involved in these two equations, I would suggest deleting the CL but showing AEZ.

[Figure]

Answer: We appreciate these thoughtful suggestions. -The units for all dependent and independent variables in Eqns (1) and (2) were added. -We added the brief description in section 2.2 to explain how we quantified the preseason water status and water regime during the rice-growing season when we were collecting data. The detailed description can be found in Table 1. -As stated in the revised manuscript, OM and AOM represent the type and amount of organic amendments added, respectively. -We followed this suggestion and changed 'CL' to 'AEZ' throughout the manuscript.

3. Results and Discussion - Suggest changing '3.3 Development of region- or country-specific emission factors ' to '3.3 Region- and country-specific emission factors' - Please make further discussion to compare the emission factors in this study with IPCC default emission factors.

Answer: We appreciate this thoughtful suggestion. Regarding the region- or country-specific emission factor, we did our best to make comparisons between our estimates and these values which are being often used in their national inventory reports. However, there were not many studies to add in the discussion for the comparison between regional emission factors with other studies. Because most countries do not have country-specific emission factors till present, we evaluated our results by the following ways: one is to use the scaling factors as shown in Table 3 to derive seasonal $CH_4$ emission as it is often presented in their national communication reports to UNFCCC, and the other one is to make indirect comparison between the national $CH_4$ inventory estimated using the 2006 IPCC guideline (Yan et al., 2009) and their national inventory reports.
* * *

---

## Author Comment (AC2) · 20 Jun 2018

Answer to Referee #2

Still some language issues, e.g. title is awkward and could deter readers/interest in the paper, many other sentences have unclear meaning and/or awkward language. Paper would definitely benefit from a thorough editing for clarity and language in general.

Answer: We appreciate and followed this suggestion. We have sent our manuscript out for the language editing service (as shown in Figure 1).

Specific issues: 1. the authors already know that ln[SOC] and OMx ln[1 + AOM] will be modeled, but we don't know where that information is from.

Answer: We appreciate the referee #2 raised this concern. In fact, the initial form of the model is an exponential relationship between emission flux and controlling factors as suggested in previous studies (Bouwman et al., 2002; Yan et al., 2005). SOC content (%) and the type and amount (t/ha) of organic amendments were factors in the above equation. It has been long recognized that CH4 flux is proportional to both SOC content and the application rate of organic amendment. As CH4 flux data do not fit a normal distribution, they fit a log-normal distribution. Thus, by fitting log-transformed flux data of CH4, the above equation was revised to the Eqn (1) in this study.? That's the reason why OM*ln(1+AOM) is modeled was added.

2. Not sure that treating pH as categorical variable is at all justified or appropriate. Why was this done? Was pH reported from the different field sites in broad categories, or measured with crude litmus paper or similar? That might be a reason, but still. . . Authors state that the relationship of pH to emissions is 'not monotonic' but from Table 2, I don't see strong enough evidence of that, especially given the questionable shoe-horning into many ns from a ranked relationship of pH with emissions simply be error? Did the authors try converting pH to concentrations of H+ ions or otherwise back-log-transforming pH values, or other logical numerical ways to treat this definitely-not-categorical variable? I don't think this statement in lines 213-215, "However, soils with a pH of 5.0-5.5 showed a much higher emission than other soils", is really true. It looks to me like soils with the lowest pH values (below 4.5) had the largest effect on CH4 emissions, and the small blips at 5- 5.5 and 7 – 7.5 are not necessarily a big deal. No other literature besides the authors' 2005 paper is cited regarding a more complicated relationship between pH and CH4 emissions to support this idea.

Answer: We appreciate the referee #2 raised this thoughtful concern. Firstly, the reason why soil pH was treated as the categorical variable is that previous findings have been suggested the existence of optimum soil pH for CH4 emission, albeit the inconsistency of reported values (Parashar et al., 1991; Wang et al., 1993). As shown in the below figure (Figure 2), soil pH values were broadly distributed across the listed range

in the text in our data set. Secondly, we found that the relationship between soil pH and CH4 flux was not monotonic. In our data set, we used pH(water) as the soil pH values for most cases. As shown in the below figure and also described in the manuscript, the largest effects of soil pH below 4.5 may not be reliable because of the limited number of observations from only two studies with large variability. The effects of soil pH above 6.0 were not significantly different from each other. Indeed, soils with a pH of 5.0-5.5 showed a much higher emission that soils with 4.5-5.0 and 5.5-6.0. Collectively, we considered the soil pH as a categorical variable which may be at least appropriate in terms of our current data sets.

3. How did the authors arrive at the weights for the organic matter additions (.2 and 1)? Not clear why this is needed or justified.

Answer: We added the explanation. There is an assumption that in cases where the amount of organic amendment is zero (i.e., no organic material added), it is the result of each type of organic material at zero application rate. By this, more data points in the analysis will have than the actual size of real observations. To ameliorate this problem, we weighted the residual of observations with organic amendments as 1 and those without as 0.2 (as the observational result was repeated five times for the five types of organic materials).

4. The authors state several times that because emissions estimates from different authors' inventory assessments, that this means the results are correct/reliable, e.g. line 70, and lines 173-175 where EDGAR estimates are similar to IPCC 2006. This is a truism, though, because doesn't EDGAR use IPCC 2006 defaults to calculate their emissions estimates?

Answer: We appreciate the referee's comment on this. In fact, the method to estimate CH4 emission from rice fields using the IPCC methodology was different among studies. For example, in Yan et al. (2009), not only the default EF used for countries where country-specific EFs were not available but also the country-specific EF derived from

various scaling factors were applied when estimating CH4 emission from global rice fields. However, in the Emission Database of Global Atmospheric Research (EDGAR), only the IPCC default EF was used (EDGAR, 2017). In addition, we have revised the sentences for clarity.

References:

Bouwman, A. F., Boumans, L. J. M. and Batjes, N. H.: Modeling global annual N2O and NO emissions from fertilized fields, Global Biogeochem. Cycles, 16(4), 28-1-28–9, doi:10.1029/2001GB001812, 2002.

EDGAR: Global Emissions EDGAR v4.3.2: part I: the three main greenhouse gases CO2, CH4 and N2O, [online] Available from: http://edgar.jrc.ec.europa.eu/overview.php?v=432_GHG&SECURE=123 (Accessed 1 November 2017), 2017.

Parashar, D.C., Rai, J., Gupta, P.K., Singh, N.: Parameters affecting methane emission from paddy fields. Indian J. Radio Space Phys, 20, 12–17, 1991.

Wang, Z.P., DeLaune, R.D., and Masscheleyn, P.H: Soil redox and pH effects on methane production in a flooded rice soil, Soil Sci. Soci. America J., 57, 382–385, 1993.

Yan, X., Ohara, T. and Akimoto, H.: Statistical modeling of global soil NOX emissions, Global Biogeochem. Cycles, 19(3), 1–15, doi:10.1029/2004GB002276, 2005.

Yan, X., Akiyama, H., Yagi, K. and Akimoto, H.: Global estimations of the inventory and mitigation potential of methane emissions from rice cultivation conducted using the 2006 Intergovernmental Panel on Climate Change guidelines, Global Biogeochem. Cycles, 23(2), doi:10.1029/2008GB003299, 2009.
* * *
[Figure]

WILEY                                    **Wiley Editing Services**

**LANGUAGE EDITING**
**CERTIFICATE**

This document certifies that the manuscript listed below was edited for proper English language, grammar, punctuation, spelling, and overall style by one or more of the highly qualified native English speaking editors at Wiley Editing Services.

**Manuscript title:**
Controlling variables and emission factors of methane from global rice fields

**Authors:**
J. Wang, H. Akiyama, K. Yagi, X. Yan

**Date Issued:**
June 19, 2018

**Certificate Verification Key:**
AA4D-0E7D-A98F-3063-C60A

This certificate may be verified at
https://secure.wileyeditingservices.com/certificate. This document
certifies that the manuscript listed above was edited for proper English
language, grammar, punctuation, spelling, and overall style. Neither
the research content nor the authors' intentions were altered in any way
during the editing process. Documents receiving this certification
should be English-ready for publication; however, the author has the
ability to accept or reject our suggestions and changes. If you have any
questions or concerns about this document or certification, please
contact help@wileyeditingservices.com.

[Figure]

**Fig. 1.** Language editing certificate

[Figure]

[Figure]

[Figure]

**Fig. 2.** Soil pH

---

## Author Response (AR1)

**Answers to Referee #1:**

General comments

Rice agriculture is an important source of atmospheric methane ($CH_4$). The estimations of $CH_4$ emission from rice fields on a national or global scale have been relatively well documented by
5  using the inventory-based methods or model-based approaches. Due to more and more field measurements of $CH_4$ emission were available from the monsoon Asian countries and the rest of the world in last ten years, the effect of various factors (management practices like water management, nitrogen (N) fertilizer use, organic input and rice varieties, etc.) on $CH_4$ emission from rice fields would be different in statistics from previous reports. However, no information is
10  available on this issue in global scale. The authors updated the dataset from monsoon Asian countries as described previously (Yan et al., 2005) to over the world (1089 measurements from 122 rice fields across the world) in this study. They reassessed the impacts of major variables controlling $CH_4$ emission from rice fields and found that water management and organic fertilizer application were the top two controlling variables. They developed the region- and country-
15  specific emission factors and also estimated the default EFs at regional and country levels. Overall, the topic of this work was very important and timely to gain an insight into $CH_4$ emission inventory, which would help to assess regional and national agricultural $CH_4$ budget with low uncertainties. Good job! The manuscript was well written too. I recommend this work to be acceptable after minor revisions for publication in Atmospheric Chemistry and Physics.

20  **Answer**: We would like to thank referee #1 for his/her positive and critical comments on our work. We are glad that referee #1 recognized the importance of our work and we would like to take the opportunity to address concerns of referee #1.

Minor comments

1. Abstract Please give more information (e.g., EFs or SFs) about the $CH_4$ emission as affected
25  by the region. In other words, the authors should pay much more attention to the regional $CH_4$ emission or emission factors (EFs) besides the management practices.

**Answer**: We followed this suggestion. Results on organic amendment and global or regional emission factors of $CH_4$ were added in the abstract.

2. Materials and Methods

30  - Please show the units for all dependent and independent variables in Eqns (1) and (2). - How to quantify the preseason water status (PW) and water regime (WR) in Eqns (1) and (2)? - What's the difference between OM and AOM in Eqn (1)? - It's hard to figure out what the climate variables are. Do the agroecological zones (AEZ) represent climates? If no climate variables were involved in these two equations, I would suggest deleting the CL but showing AEZ.

35 **Answer**: We appreciate these thoughtful suggestions. -The units for all dependent and independent variables in Eqns (1) and (2) were added. -We added the brief description in the section 2.2 to explain how we quantified the preseason water status and water regime during the rice-growing season when we were collecting data. The detailed description can be found in Table 1. -As stated in the revised manuscript, OM and AOM represent the type and amount of

40 organic amendments added, respectively. -We followed this suggestion and changed 'CL' to 'AEZ' throughout the manuscript.

3. Results and Discussion - Suggest changing '3.3 Development of region- or country-specific emission factors ' to '3.3 Region- and country-specific emission factors' - Please make further discussion to compare the emission factors in this study with IPCC default emission factors.

45 **Answer**: We appreciate this thoughtful suggestion. Regarding the region- or country-specific emission factor, we did our best to make comparisons between our estimates and these values which are being often used in their national inventory reports. However, there were not many studies to add in discussion for the comparison between reginal emission factors with other studies. Because most countries do not have country-specific emission factors till present, we

50 evaluated our results by the following ways: one is to use the scaling factors as shown in Table 3 to derive seasonal CH4 emission as it is often presented in their national communication reports to UNFCCC, and the other one is to make indirect comparison between the national CH4 inventory estimated using the 2006 IPCC guideline (Yan et al., 2009) and their national inventory reports.

55

**Answer to Referee #2**

Still some language issues, e.g. title is awkward and could deter readers/interest in the paper, many other sentences have unclear meaning and/or awkward language. Paper would definitely benefit from a thorough editing for clarity and language in general.

60 **Answer**: We appreciate and followed this suggestion. We have sent our manuscript for language editing service (as shown in Figure 1).

Specific issues:

1. the authors already know that ln[SOC] and OMx ln[1 + AOM] will be modeled, but we don't know where that information is from.

65 **Answer**: We appreciate the referee #2 raised this concern. In fact, the initial form of the model is an exponential relationship between emission flux and controlling factors: $\text{flux} = e^{constant + \sum_i factor(i)}$, as suggested in previous studies (Bouwman et al., 2002; Yan et al., 2005). The SOC content (%) and the type and amount (t/ha) of organic amendments were factors in the above equation. It has been long recognized that CH4 flux is proportional to both SOC content

and the application rate of organic amendment. As CH4 flux data do not fit a normal distribution, they fit a log-normal distribution. Thus, by fitting log-transformed flux data of CH4, the above equation was revised to the Eqn (1) in this study.? That's the reason why OM*ln(1+AOM) is modeled was added.

2. Not sure that treating pH as categorical variable is at all justified or appropriate. Why was this done? Was pH reported from the different field sites in broad categories, or measured with crude litmus paper or similar? That might be a reason, but still. . . Authors state that the relationship of pH to emissions is 'not monotonic' but from Table 2, I don't see strong enough evidence of that, especially given the questionable shoe-horning into many ns from a ranked relationship of pH with emissions simply be error? Did the authors try converting pH to concentrations of H+ ions or otherwise back-log-transforming pH values, or other logical numerical ways to treat this definitely-not-categorical variable? I don't think this statement in lines 213-215, "However, soils with a pH of 5.0-5.5 showed a much higher emission than other soils", is really true. It looks to me like soils with the lowest pH values (below 4.5) had the largest effect on CH4 emissions, and the small blips at 5- 5.5 and 7 – 7.5 are not necessarily a big deal. No other literature besides the authors' 2005 paper is cited regarding a more complicated relationship between pH and CH4 emissions to support this idea.

**Answer**: We appreciate the referee #2 raised this thoughtful concern. Firstly, the reason why soil pH was treated as categorical variable is that previous findings have been suggested the existence of optimum soil pH for CH4 emission, albeit the inconsistency of reported values (Parashar et al., 1991; Wang et al., 1993). As shown in the below figure (Figure 2), soil pH values were broadly distributed across the listed range in the text in our data set. Secondly, we found that the relationship between soil pH and CH4 flux was not monotonic. In our data set, we used pH(water) as the soil pH values for most cases. As shown in the below figure and also described in the manuscript, the largest effects of soil pH below 4.5 may not be reliable because of the limited number of observations from only two studies with large variability. The effects of soil pH above 6.0 were not significantly different from each other. Indeed, soils with a pH of 5.0-5.5 showed a much higher emission that soils with 4.5-5.0 and 5.5-6.0. Collectively, we considered the soil pH as a categorical variable which may be at least appropriate in terms of our current data sets.

3. How did the authors arrive at the weights for the organic matter additions (.2 and 1)? Not clear why this is needed or justified.

**Answer**: We added explanation. There is an assumption that in cases where the amount of organic amendment is zero (i.e., no organic material added), it is the result of each type of organic material at zero application rate. By this, more data points in the analysis will have than the actual size of real observations. To ameliorate this problem, we weighted the residual of observations with organic amendment as 1 and those without as 0.2 (as the observational result was repeated five times for the five types of organic materials).

4. The authors state several times that because emissions estimates from different authors' inventory assessments, that this means the results are correct/reliable, e.g. line 70, and lines 110 173-175 where EDGAR estimates are similar to IPCC 2006. This is a truism, though, because doesn't EDGAR use IPCC 2006 defaults to calculate their emissions estimates?

**Answer**: We appreciate the referee's comment on this. In fact, the method to estimate CH4 emission from rice fields using the IPCC methodology were different among studies. For example, in Yan et al. (2009), not only the default EF used for countries where country-specific 115 EFs were not available but also the country-specific EF derived from various scaling factors were applied when estimating CH4 emission from global rice fields. However, in the Emission Database of Global Atmospheric Research (EDGAR), only the IPCC default EF was used (EDGAR, 2017). In addition, we have revised the sentences for clarify.

Figure 1

**WILEY**

**Wiley Editing Services**

**LANGUAGE EDITING**

**CERTIFICATE**

This document certifies that the manuscript listed below was edited for proper English language, grammar, punctuation, spelling, and overall style by one or more of the highly qualified native English speaking editors at Wiley Editing Services.

**Manuscript title:**

Controlling variables and emission factors of methane from global rice fields

**Authors:**

J. Wang, H. Akiyama, K. Yagi, X. Yan

**Date Issued:**

June 19, 2018

**Certificate Verification Key:**

AA4D-0E7D-A98F-3063-C60A

This certificate may be verified at https://secure.wileyeditingservices.com/certificate. This document certifies that the manuscript listed above was edited for proper English language, grammar, punctuation, spelling, and overall style. Neither the research content nor the authors' intentions were altered in any way during the editing process. Documents receiving this certification should be English-ready for publication; however, the author has the ability to accept or reject our suggestions and changes. If you have any questions or concerns about this document or certification, please contact help@wileyeditingservices.com.

[Figure]

Wiley Publishing Services is a service of Wiley Publishing. Wiley's Scientific, Technical, Medical, and Scholarly (STMS) business serves the world's research and scholarly communities, and is the largest publisher for professional and scholarly societies. Wiley is committed to providing high quality services for researchers. To find out more about Wiley Editing Services, visit wileyeditingservices.com. To learn more about our other author services provided by Wiley Publishing, visit authorservices.wiley.com.

140   Figure 2

[revised manuscript text omitted]